# Information Flow in Biological Networks for Color Vision

**DOI:** 10.3390/e24101442

**Published:** 2022-10-10

**Authors:** Jesús Malo

**Affiliations:** Image Processing Lab, Universitat de Valencia, 46980 Valencia, Spain; jesus.malo@uv.es; Tel.: +34-963-544-099

**Keywords:** chromatic information, color appearance networks, efficient coding hypothesis, total correlation, mutual information, Gaussianization

## Abstract

Biological neural networks for color vision (also known as color appearance models) consist of a cascade of linear + nonlinear layers that modify the linear measurements at the retinal photo-receptors leading to an internal (nonlinear) representation of color that correlates with psychophysical experience. The basic layers of these networks include: (1) chromatic adaptation (normalization of the mean and covariance of the color manifold); (2) change to opponent color channels (PCA-like rotation in the color space); and (3) saturating nonlinearities to obtain perceptually Euclidean color representations (similar to dimension-wise equalization). The *Efficient Coding Hypothesis* argues that these transforms should emerge from information-theoretic goals. In case this hypothesis holds in color vision, the question is *what is the coding gain due to the different layers of the color appearance networks*? In this work, a representative family of color appearance models is analyzed in terms of how the redundancy among the chromatic components is modified along the network and how much information is transferred from the input data to the noisy response. The proposed analysis is performed using data and methods that were not available before: (1) new colorimetrically calibrated scenes in different CIE illuminations for the proper evaluation of chromatic adaptation; and (2) new statistical tools to estimate (multivariate) information-theoretic quantities between multidimensional sets based on Gaussianization. The results confirm that the efficient coding hypothesis holds for current color vision models, and identify the psychophysical mechanisms critically responsible for gains in information transference: opponent channels and their nonlinear nature are more important than chromatic adaptation at the retina.

## 1. Introduction

Biological vision is relevant to manage visual data because natural visual systems evolved to develop efficient representations of visual features that may be an inspiration for artificial systems. Examples include (1) the equivalence between the spatio-spectral sensitivity of receptive fields in natural neural systems and those emerging from information maximization and optimal matching [1], or the filters found by maximizing classification performance [2], and (2) the importance of human-like spatio-chromatic representations in image coding algorithms [3,4].

Conversely, the quantitative tools of statistical learning are key to propose principled theories in visual neuroscience. For instance, the classical *efficient coding hypothesis* argues that the organization of biological sensors comes from the optimization of information-theoretic goals [5,6]. The conventional approach to check this hypothesis is *from-statistics-to-perception*: i.e., deriving the biological behavior from statistical arguments. In color vision, this includes the derivation of opponent channels from principal components [7], and the derivation of the frequency bandwidth of color channels [8,9] nonlinearities of opponent channels [10,11], and even the reproduction of color illusions [12,13], from information maximization or error minimization arguments. However, there is an alternative way to check the hypothesis: *from-perception-to-statistics* [14,15,16]. In this case, perceptually meaningful models which have not been statistically optimized are shown to have a statistically efficient behavior.

In this work, we take this alternative approach (*from-perception-to-statistics*) for color vision models. Here, we analyze the communication efficiency of standard color vision models (that already describe a wide range of color vision psychophysics) using multivariate information-theoretic measurements. This analysis is interesting for data science and computer vision because the building blocks of color appearance models have received statistical interpretation: adaptation is related to manifold alignment [17], opponency is related to the principal components of color data [7], and the nonlinearities of the opponent channels are related to histogram equalization [10,11,12]. In case the efficient coding hypothesis holds for color vision networks, the question is *what is the coding gain due to the different layers of the color appearance networks?*

A quantitative response to that question is interesting for data science to select which process should be addressed first for the efficient management of color data. The *perception-to-statistics* approach has been previously applied to texture perception [14,15] and to models that involve spatial transforms [16], but it is original in purely color vision: note that previous information-theoretic analysis of color vision (e.g., Ref. [18] and the references therein) were focused on the amount of information from an image which can be obtained from the corresponding scene under a different illumination and not on the efficiency of the system to transmit generic color information (irrespective of the illumination) as done herein. The work presented here is an extended version of the preliminary results presented in an oral communication [19].

## 2. Statistical Interpretation of Building Blocks of Color Vision

The basic elements of biological color vision are:

**(1) Linear integration of spectral irradiance at the retina** by three sensors tuned to *Long*, *Medium* and *Short* wavelengths, which are commonly referred to as LMS sensors [20]. In every natural or artificial system, this initial linear stage is the necessary transduction from electromagnetic energy to the first numerical representation of color data. Here, we will start by expressing colorimetrically calibrated images in LMS tristimulus values via the Stockman and Sharpe fundamentals [21].

**(2) Nonlinear adaptation at the retina** adjusts the sensitivity (or gain) of LMS sensors to the illumination of the scene. For instance, classical von Kries chromatic adaptation normalizes the sensors by the responses of what is considered to be *white* in the scene [22]. From a statistical point of view, the role of chromatic adaptation is the same as manifold alignment in machine learning to make the interpretation of the data easier in changing environments (in this case, environments with different illuminations). According to this interpretation, generalizations of the von Kries transform have been proposed, such as, for instance, trying to make first and second moments of the different color manifolds equal [17,23], or using higher-order equalization transforms for the different datasets and making their dimensions equal in the canonical domain. The latter higher-order methods may be linear [1] or nonlinear [11,12]. Here, we will explore the behavior of classical von Kries adaptation [22], and the adaptation through the equalization of mean and covariance, which will be referred to as the Webster–Clifford approach following [17,23]. More sophisticated nonlinear equalization techniques such as the sequential principal curves analysis (SPCA) [11,12] will be used as a convenient statistical benchmark.

**(3) Linear opponent channels in ganglion cells and beyond**. The change from a color representation mediated by sensors with *all-positive* (physically realizable) sensitivities, as the LMS sensors at the retina, to color representations in *opponent channels* is obtained via the linear recombination of the LMS signals. This recombination leads to an achromatic channel and two red-green and yellow-blue chromatic channels [20,22,24]. As such, neural computation allows to obtain sensors with *opponent* (positive-and-negative) spectral sensitivities that are not easy to implement physically. Spectral sensitivities which are effectively opponents are found at different layers along the neural pathway: at the ganglion cells, the lateral geniculate nucleus, and the visual cortex [25]. This linear change of color representation has been statistically interpreted as the identification of the principal components of the color manifold [7]. Here, we will use the classical Jameson and Hurvich transform to opponent channels [24], which is not based on image statistics, but on color-matching experiments. These opponent channels are also called *achromatic*, *tritanopic,* and *deuteranopic* (ATD) due to their relation with dichromatic vision [20,26].

**(4) Nonlinearities of opponent channels**. Nonuniform color discrimination thresholds [27,28] imply that the saturation of the Weber law occurring in the achromatic channel [22,29] also appears in the chromatic channels [20]. This nonlinearity of the opponent channels has been explained as the necessary transform to obtain equalized PDFs in the responses [10]. This nonlinearity can be optimized for PDF equalization or error minimization after uniform quantization. The authors referred to this nonlinearity as the pleistochrome transform [10]. This dimension-wise equalization concept was generalized to multivariate scenarios through principal curves (the above mentioned SPCA) [11,12]. Here, the simpler (univariate) pleistochrome transform will be compared to the more general SPCA transform.

The sequence of the building blocks considered above fits into the current deep-network paradigm [30] because it can be implemented as a cascade of two *linear+nonlinear* layers performing *spectral integration + adaptation* followed by *opponency + saturation*: (1)
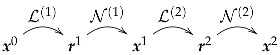


In this architecture, $x^{0}$ is the spectral irradiance at the photo-receptors, $r^{1}$ is the vector of linear LMS responses, $x^{1}$ is the vector of nonlinearly adapted LMS responses, $r^{2}$ is the vector of linearly recombined ATD responses, and $x^{2}$ is the vector of nonlinear ATD responses. The goal of the work was to measure how much information from the linear LMS stage is transmitted to the other layers of the network. As such, we check whether information transmission can be a sensible organization principle for these natural systems which are the critical layers of the network.

## 3. Materials and Methods

### 3.1. Colorimetrically Calibrated Database

The IPL color image database [1,11,12] is well suited to study color adaptation because its controlled illumination under CIE A and CIE D65 allows us to know the white point in each scene. This implies that chromatic adaptation transforms will not require extra approximations such as the gray-world assumption. The acquisition of the controlled images and resulting data is illustrated in Figure 1.

### 3.2. Quantification of the Information Flow

The information flow along the considered networks can be measured in direct and indirect ways. The direct measure consists of estimating the information about the input natural colors $x^{0}$ contained at the color representation (or response) at the *i*-th layer of the network, $x^{i}$: the *Mutual Information*$I(x^{0},x^{i})$. Of course, the information transferred from the input can also be measured for the stages after the linear transforms, $I(x^{0},r^{i})$. Given samples of the color signals at the different stages (either $r^{i}$ or $x^{i}$), the same numerical technique will be applied to estimate the corresponding mutual information *I*. Therefore, below we will give expressions for $x^{i}$ which will also be applicable for $r^{i}$.

These direct measures, *I*, can be put in terms of other (indirect) magnitudes. For illustration purposes, following [16], one can compare the efficiency of the representation in the *i*-th layer assuming that it performs a certain deterministic transform, $f^{i}\left(\cdot \right)$, corrupted by noise due to the non-ideal nature of the sensors: $x^{i}=f^{i}\left(x^{0}\right)+n^{i}$. In this setting, from the definition of *I* in terms of the joint and the conditional entropies (chapter 2 in [31]) leads to: $I\left(x^{0},x^{i}\right)=h\left(x^{i}\right)-h\left(x^{i}|x^{0}\right)=h\left(x^{i}\right)-h\left(n^{i}\right)$, because the uncertainty in the *i*-th layer given the input just comes from the noise $n^{i}$. Then, from the definition of *Total Correlation*, *T*, Refs. [32,33] that describe the redundancy between the coordinates, $x_{j}$, of a *d*-dimensional vector, $T\left(x\right)=\sum_{j=1}^{d}h\left(x_{j}\right)-h\left(x\right)$, the information transferred up to the *i*-th layer is just [16]:(2)$$I\left(x^{0},x^{i}\right)=\sum_{j=1}^{d}h\left(x_{j}^{i}\right)-T\left(x^{i}\right)-h\left(n^{i}\right)$$
where the superindex refers to the *i*-th layer and the subindex *j* refers to the individual neurons within that layer.

The right-hand side of Equation (Equation 2) implies that the transferred information can be increased in different ways: (1) by increasing the entropy of the coefficients of the response, which is limited by energy constraints that prevent arbitrarily large gains to increase the variance; (2) by reducing the noise at that layer, which is also limited by the finite quality of the sensors; and finally (3) by reducing the redundancy, *T*, between the sensors in the representation. Therefore, for sensors of fixed signal/noise quality, indirect measures of efficiency include the redundancy, *T*, within each layer along the networks, and the differential entropy, *h*, of the signal at each layer of the networks.

Equation (Equation 2) introduced in [16] clarifies the statements made in [34]: representations that minimize the redundancy, *T*, and maximize the entropy, *h*, are better for signal representation because independent components lead to factorial codes that maximize the use of the channel (maximize the transferred information), and maximum entropy representations imply that the signal accepts more noise without significant information loss.

By definition, noiseless and invertible representations preserve all the information from the input. Therefore, the measurement of the transmitted information only makes sense for noisy sensors (which is the case of actual physiological mechanisms). According to this, just for illustrative purposes as in [16], in our experiments measuring *I*, every considered sensor (or representation) was subject to Gaussian noise whose standard deviation was 5% of the total deviation of the response. We computed *I* between these noisy representations and the input (assuming a negligible noise in the input, with a standard deviation of 0.05% of the total deviation). The *I* between the noiseless input and the negligible-noise input will set a convenient reference for the maximum available information.

Finally, the Kullback–Leibler divergence, *KLD* or $D_{KL}$, which measures the lack of correspondence between data distributions [31], is a convenient information-theoretic measure to assess the match between the color-compensated sets using different chromatic adaptation mechanisms. Lower divergences indicate better adaptation.

In summary, *I*, *T*, *h*, and $D_{KL}$ are appropriate measures to assess the information flow and adaptation in color appearance networks. However, the estimation of these (multivariate) quantities directly from the color samples injected into the networks is not straightforward: the naive use of the direct definitions implies the estimation of multivariate PDFs and this would introduce substantial bias in the results. This estimation problem is addressed in the next section.

### 3.3. Total Correlation, Mutual Information, and Kullback–Leibler Divergence from Gaussianization

In this work, we solve the problem of estimating multivariate information-theoretic quantities using a novel estimator of *T* which only relies on easier (univariate) density estimations: the rotation-based iterative Gaussianization (RBIG) [35,36].

The RBIG transform, $x^{\prime }=G_{x}\left(x\right)$, is a cascade of nonlinear+linear iterations, or the sequential application of two operations: nonlinear marginal Gaussianizations, Ψ, and linear rotations, R: (3)
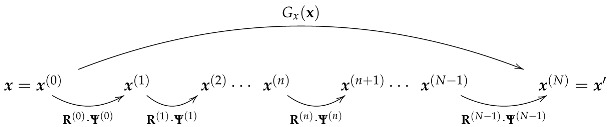

where *n* refers to the step in the sequence, and convergence takes *N* steps, n=0,…,N−1.

This invertible iterative procedure in Equation (Equation 3) is able to transform any input PDF into a zero-mean unit-covariance multivariate Gaussian even if the chosen rotations are random [35]. This ability to completely remove the structure of any PDF is useful to estimate the *T* of arbitrary vectors x: as the redundancy of a Gaussianized signal is zero, T(x) corresponds to the sum of the individual variations that take place along the iterations of RBIG while Gaussianizing x. Interestingly, the individual variation in each RBIG iteration only depends on (easy to compute) univariate entropies [36]:(4)$$T\left(x\right)=\sum_{n=0}^{N-1}\Delta T^{\left(n\right)}=\frac{(N-1)d}{2}\log\left(2\pi e\right)-\sum_{n=1}^{N}\sum_{j=1}^{d}h\left(x_{j}^{\left(n\right)}\right)$$
where the (hard) multivariate problem was reduced to the computation of many (easy to compute) univariate entropies $h(x_{j}^{\left(n\right)})$.

The entropy of the multivariate signal at the *i*-th layer, $h(x^{i})$, can be obtained from $T(x^{i})$, and univariate entropy estimations from the aforementioned definition of *T*:(5)$$h\left(x^{i}\right)=-T\left(x^{i}\right)+\sum_{j=1}^{d}h\left(x_{j}^{i}\right)$$

The information shared by multidimensional datasets, *I*, is just the remaining total correlation within the variables once they have been separately Gaussianized [36,37]:(6)$$I\left(x,y\right)=T(\left[G_{x}\left(x\right),G_{y}\left(y\right)\right]),$$
where $G_{x}\left(\cdot \right)$ and $G_{y}\left(\cdot \right)$ are the Gaussianization transforms learnt for the random variables **x** and **y**, and *T* in Equation (Equation 6) can be reduced to univariate operations using Equation (Equation 4).

Similarly, the *KLD* between two multidimensional variables depends on the redundancy and the negentropy of one of the variables when it has been Gaussianized using the Gaussianization transform learnt from the other [36]:(7)$$D_{\mathrm{KL}}\left(x|y\right)=T\left(z\right)+\sum_{j=1}^{d}D_{\mathrm{KL}}\left(p_{z_{j}}\left(z_{j}\right)|N\left(0,1\right)\right)$$
where $z=G_{x}\left(y\right)$ is the variable **y** transformed according to the Gaussianization mapping for **x**.

In summary, according to Equations (Equation 4)–(Equation 7), all the hard-to-compute multivariate measures (*T*, *h*, *I*, $D_{KL}$) can be reduced to the computation of univariate quantities thanks to RBIG.

We also computed *T* and *I* via the Kozachenko–Leonenko estimator [38] used in other studies on color data [39]. However, note that this alternative estimator cannot be used to compute the *KLD*.

## 4. Experiments and Results

Here, we studied the communication efficiency of three illustrative families of color vision models:**Physiological networks:** Cascades of physiologically meaningful linear + nonlinear layers:(i)LMS sensors [21] + von Kries or Webster–Clifford adaptation [17], followed by(ii)Jameson and Hurvich linear opponent channels [24] + pleistochrome saturations [10].**Psychophysical networks:** Standard color appearance models that are made of the same ingredients, including(i)The classical CIE L${}^{*}$a${}^{*}$b${}^{*}$ model [40],(ii)The LLAB model [22,41], and(iii)The more recent CIECAM02 [42,43].**Statistical models:** Multivariate equalization based on principal curves, SPCA, was used to explain color vision [11,12], and this will also be considered (with or without classical chromatic adaptation transforms) as a convenient statistically based benchmark.

We used the following parameters in the above models. In the von Kries adaptation, the corresponding *white points* were computed by the mean of the color samples obtained under the different illuminations. The same was performed in the Webster–Clifford approach, which also used the sample covariance matrices for manifold alignment. The pleistochrome saturation functions were approximated by exponential functions. All the psychophysical networks also used the mean of the color samples for the different illuminations to obtain the *white points*. This is the only parameter in CIE L${}^{*}$a${}^{*}$b${}^{*}$. However, LLAB and CIECAM02 require extra parameters. In LLAB, we took the adaptation and inductions parameters listed in [22] for a default *average surround*. In CIECAM02, we also assumed an *average surround* (with the corresponding consequence in adaptation, impact of surround, and induction parameters [43]). Moreover, we took the *adapting field*, the *background*, and the *white point* from the mean of the colors. Given the luminances of these mean colors, the degree of adaptation in CIECAM02 was approximately 0.9 in the [0,1] range. Finally, in the statistical SPCA, we used the parameters for information maximization in color manifolds reported in [11], which imply relatively stiff principal curves but adaptive histogram equalization (or generalized pleistochrome) at different points of these curves.

The physiological and psychophysical networks are implemented in Colorlab [44] available here (https://isp.uv.es/code/visioncolor/colorlab.html, accessed on 5 October 2022) and the SPCA algorithm [11,12] is available here (https://isp.uv.es/spca.html, accessed on 5 October 2022).

### 4.1. Visualization of the Color Manifolds

First, we show how the color manifolds of natural scenes under the CIE D65 and CIE A illuminations change through the layers of the considered networks. Information measures are related to the volume and density of the data distribution [31]. As a result, the visualization of the shape of the manifolds is important to understand the geometric effect of the considered transforms, and hence, their impact on the measures. Specifically, Figure 2 shows the separate effect of the elements of the physiological networks (opponency and nonlinearities of opponent channels without and with different chromatic adaptations). Figure 3 shows the effect of different (psychophysical) color appearance models once the corresponding chromatic adaptation has been performed and after the nonlinear opponent responses have been obtained. Finally, Figure 4 shows the result of the nonlinear equalization SPCA when it is applied to nonaligned manifolds or to manifolds aligned according to different color adaptation transforms.

### 4.2. Quantification of the Information Flow

We provide indirect measures of efficiency, such as the redundancy, *T*, within each layer along the networks, and the differential entropy, *h*, of the signal at each layer of the networks (Table 1 and Table 2, respectively). We also provide a direct measure of efficiency: the mutual information between the noiseless input and the noisy response, *I*, at each layer of the network (Table 3).

These magnitudes were estimated from the responses of the models to test sets of randomly chosen color samples from the IPL database shown in Figure 1. The parameters of the models that require statistical training (namely the Webster–Clifford adaptation, the pleistochrome nonlinearity, and SPCA) were trained over a set with $2\times 10^{6}$ natural color samples. Then, $4\times 10^{4}$ color samples not included in the training set were transformed using the perceptual models and the purely statistical SPCA. Ten independent estimations of the quantities were performed using random subsets of 80% of the test set. All tables shown were estimated using the RBIG (code in https://isp.uv.es/RBIG4IT.htm, accessed on 5 October 2022). Additionally, entropy and mutual information were also estimated using the Kozachenko–Leonenko procedure leading to qualitatively similar results (tables equivalent to Table 2 and Table 4, not shown).

Finally, in Table 5, we report the *KLD* that measures the correspondence between the color sets corresponding to D65 illumination and A illumination after chromatic adaptation (or white balance).

The tables have to be interpreted according to the following reference values:Total correlation in Table 1 should be interpreted in light of Equation (Equation 2) where the transmitted information is maximized only if redundancy is completely removed: optimal means *T* = 0.The differential entropy, *h*, depends on arbitrary changes of scale (or units) of the response. It can be even negative for PDFs of small support (chapt. 17 in [31]). Thus, for fair comparison, all *h* values in Table 2 were computed after linearly re-scaling the signals to be inscribed in the same 3D cube of size *S*. As such, the *h* values do describe how uniform the distributions are in the common support (in our case, we chose *S* = 10). As a useful reference, the upper bound for *h* is achieved by the uniform distribution, which is $h_{\mathrm{unif}}=d\;\log_{2}S=3\;\log_{2}10=9.97$ bits.As stated in Section 3.2, the information shared by the noiseless input, $x^{0}$, and the negligible-noise input, $x_{☆}^{0}$, is a convenient reference because other (more noisy) layers will share less information with $x^{0}$. Assuming, as indicated in Section 3.2, that noise in LMS with 0.05% deviation is negligible, the empirical RBIG computation of $I(x^{0},x_{☆}^{0})$ on natural colors in LMS gives 14.1±0.1 bits. The values of *I* in Table 3 should be compared to that upper bound.In Table 4, lower divergences indicate a better match between the color compensated sets, so the optimal value would be $D_{KL}=0$.

In all tables, the three best results are highlighted in blue.

## 5. Discussion

The visualization of the manifold changes through the transforms of the considered models confirms their statistical interpretation outlined in the introduction. Adaptation aligns the data obtained under different acquisition conditions: compare the unaligned sets in blue and red in the plots of the left column of Figure 2 (no adaptation) with all the other cases where both sets overlap. Linear opponent channels rotate the input LMS representation following the axes of the data: this rotation is particularly clear in the transition from the first row to the second row in Figure 2, which shows rotations of the sets consistent with the principal component analysis interpretation of opponent channels [7]. Finally, the nonlinear nature of the opponent channels equalizes the responses: note in the bottom row of Figure 2 and Figure 3 how the biological networks make the different dimensions comparable in size. The nonlinearities equalize the responses as opposed to the linear representations where the luminance has substantially bigger energy and the PDF is more nonuniform. This is consistent with the equalization goal proposed in [10,11,45].

While data in the input representation are highly correlated due to the overlap of the LMS sensitivities (strong alignment along the diagonal of the domain in the top row of Figure 2 and Figure 3), the redundancy between the responses clearly reduces at later stages. This is quantitatively confirmed by the reduction in *T* along the layers of the networks in Table 1. On the other hand, visualizations suggest that data are progressively equalized along the networks and this is quantitatively confirmed by the progressive increase in the differential entropy along the columns of Table 2. As a result of the progressive independence and equalization, the amount of information about the input available at the different representations increases along the columns of Table 3.

It is remarkable that psychophysically-tuned models such as CIE-Lab and CIECAM have similar or better information transmission performance than an unsupervised learning method, SPCA, specifically trained for information maximization. While the emergence of perceptual nonlinearities, adaptation, and aftereffects from SPCA [11,12] is a confirmation of the efficient coding hypothesis in the classical *from-statistics-to-perception* direction, the quantitative efficiency of CIE color appearance models presented here is a confirmation in the opposite direction (*from-perception-to-statistics*). The results presented here complement in the purely chromatic modality previous results of this alternative approach to check Barlow’s hypothesis [14,15,16].

Beyond this confirmation, our results enable the quantification of the gain in information transference due to the specific layers of the network: retinal adaptation, transform to opponent channels, and the saturation nonlinearities of the opponent channels. The average results of ΔI are given in Table 5. These gains, ΔI, imply that *opponency* is the most relevant feature of color vision to favor the efficient information transmission, followed by the *saturating response* of the opponent channels. These processes are way more important than *chromatic adaptation*. One may argue that the goal of chromatic adaptation is making scene interpretation more robust and not merely improving the information capacity of the visual pathway.

Note also that *KLD* results in Table 4 show that SPCA (designed for information maximization) actually achieves a lower adaptation performance than the simple von Kries or Webster–Clifford procedures in the LMS space. This, together with the small impact of chromatic adaptation in information transmission, suggest that adaptation should be explained by a different principle. The small effect of chromatic adaptation in information theoretic terms was also identified in a previous analysis of biological spatio-chromatic vision models [16]: von Kries did not reduce redundancy and the chromatically adapted domain with the same amount of noise shared less information with the original signal than the linear LMS space. In that study [16], the benefits of chromatic adaptation in information transmission were only apparent, if any, after the consideration of spatial transforms.

The communication efficiency analysis performed herein following the *perception-to-statistics* logic is different from the work of Foster et al., which was also concerned about the use of accurate information-theoretic measures in color vision [18]. Note that their work is mainly focused on determining the number of discriminable colors/surfaces under different illumination conditions [46,47,48], which is related to the amount of color information in a scene that can be extracted from color measurements under other illumination [18,39]. These problems are related to entropy and mutual-information measures, but they do not quantify the information transference through the visual pathway (mutual information between layers and redundancy within layers). As an example, in [47,49], the redundancy is only considered because of its impact on the available information in the color compensation context, not as a measure of information transmission of the visual system.

In more general scenarios (where perception is complemented with action), it has been suggested that information flow is not solely a function of the scenes to be encoded, but it can be maximized through *active* interaction between the system and the environment [50], e.g., through saccades, foveation, or adaptive saliency. This (more general) perception–action loop is beyond the scope of our work which is restricted to bottom–up feed-forward networks. However, from the technical point of view, the estimation of information-theoretic quantities using Gaussianization can relax the severe quantization and Gaussian assumptions that had to be used in [50]. Similarly, following [51], where estimations based on Gaussianization were used to quantify connectivity between nodes in neural networks, RBIG could be used to measure normalized mutual information which is the core of successful methods to identify actual relations in noisy networks [52].

## 6. Conclusions

The results show that biological architectures that mediate color vision are quite efficient in information theoretic terms: they reduce approximately 75% of the redundancy present at the input linear responses (Table 1 shows that the redundancy of the raw signal (input linear LMS) is 5.71 bits while the redundancy in the inner representation of psychophysical models is in the range of [1,2] bits, which represents reductions of [65,82]% of the redundancy present in the input). As a result, while noisy sensors at the input representation would only retain approximately 35% of the chromatic information (Table 2 shows that LMS sensors with noise of 5% deviation would retain 5.0 bits compared to the ideal 14.1 bits assuming negligible noise of 0.05% deviation), sensors with the same amount of noise at the inner representations retain approximately 65% of the information. The inner representation of the psychophysical models with 5% noise shown in Table 2 retain approximately [7.8, 9.8] bits from the input. This is approximately 65% of the maximum available information, 14.1 bits). The results of biological networks (not explicitly optimized for statistical goals) are on par with those of non-linear equalization methods. In terms of communication efficiency, the most relevant transform in color vision is the consideration of opponent channels followed by the nonlinear response of the opponent channels. On the contrary, the impact of adaptation to improve the transmission is almost negligible (see Table 5). The small impact of chromatic adaptation in information transmission is consistent with the previous analysis of the von Kries transform in more general (spatio-chromatic) vision models [16].

From the theoretical neuroscience perspective, these results confirm the *Efficient Coding Hypothesis* for human color vision in the *perception-to-statistics* direction: statistically agnostic color vision models such CIE L${}^{*}$a${}^{*}$b${}^{*}$ and CIECAM02 (only based on psychophysics) are remarkably efficient in transmitting natural colors. Moreover, for the data science community, these results rank the relevance of color vision features in terms of their impact in optimal color information transmission: linear decorrelation and nonlinear equalization are more important than manifold alignment (or white balance).

## Figures and Tables

**Figure 1 entropy-24-01442-f001:**
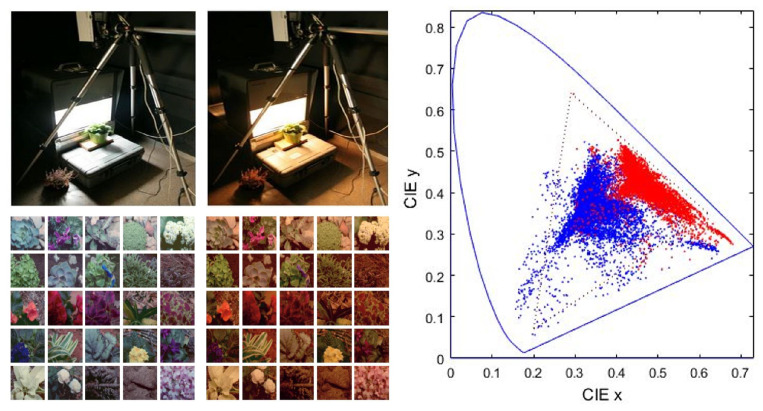
Natural colors used in the experiments. (**Left**): experimental setting to record the natural scenes under calibrated illumination (CIE D65 and CIE A spectra) and representative corresponding scenes. (**Right**): color measurements in the CIE xy diagram. Blue and red dots represent the colors under D65 (white) and A (yellowish) illuminations.

**Figure 2 entropy-24-01442-f002:**
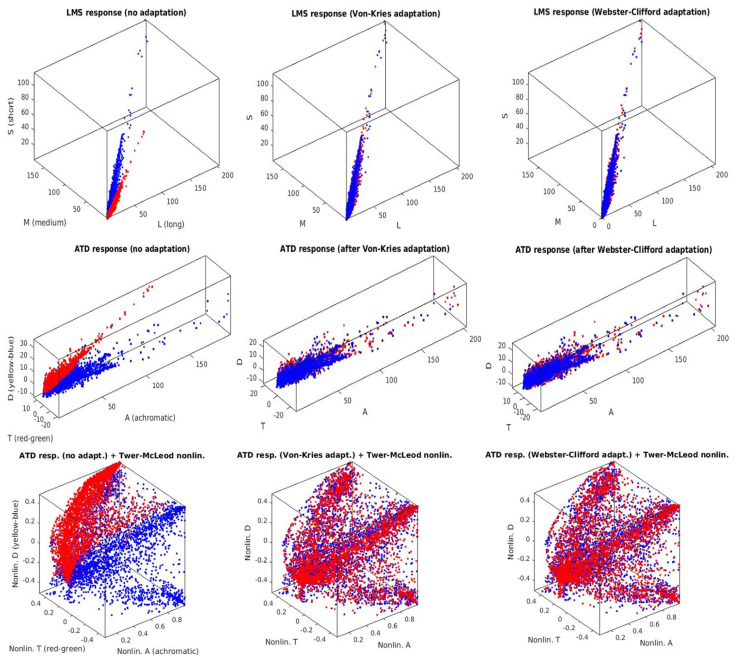
Manifold changes through physiological networks. (**Top Row**): Retinal responses of LMS cones with no adaptation (left) and with different retinal adaptation schemes (center and right). (**Center Row**): Linear opponent ATD responses (recombination after the retina) with no adaptation (left) and different adaptation schemes. (**Bottom Row**): Nonlinear opponent ATD responses after dimension-wise PDF equalization (pleistochrome transform). Red and blue dots represent the samples under the A and D65 illuminations.

**Figure 3 entropy-24-01442-f003:**
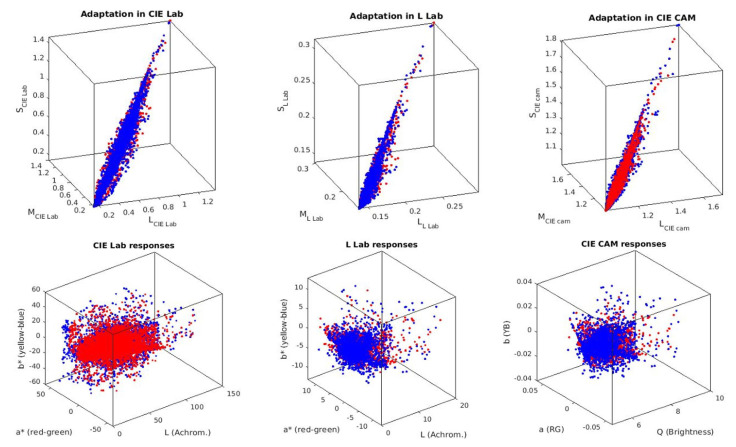
Manifold changes through psychophysical color appearance models. (**Top**): Responses of sensors of standard models tuned to long, medium, and short wavelengths. All these representations already include divisive adaptation schemes for manifold alignment. (**Bottom**): Nonlinear opponent responses in the different color appearance models. Red and blue dots represent the samples under the A and D65 illuminations.

**Figure 4 entropy-24-01442-f004:**
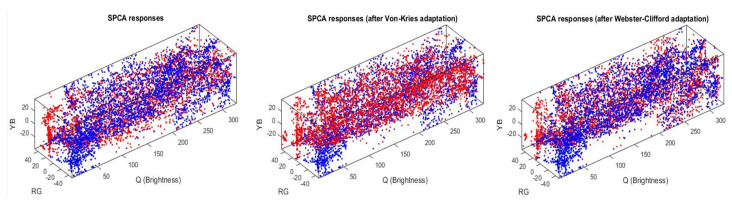
Manifold equalization using nonlinear independent component analysis (sequential principal curves, SPCA). (**Left**): Responses of the SPCA sensors from non-aligned manifolds. (**Center**): SPCA responses from von Kries adapted measurements. (**Right**): SPCA responses from Webster–Clifford adapted colors. Red and blue dots represent the samples under the A and D65 illuminations.

**Table 1 entropy-24-01442-t001:** Intra-layer Total Correlation through the networks (in bits). Optimal systems (Equation (Equation 2)) should have *T* = 0 bits.

Physiol. Models	No Adaptation	Von Kries	Webster–Clifford
Linear LMS input	5.71 ± 0.04	6.26 ± 0.02	6.29 ± 0.03
Linear ATD channels	3.23 ± 0.03	2.21 ± 0.03	2.30 ± 0.05
Pleistochrome	3.26 ± 0.04	2.22 ± 0.04	2.33 ± 0.05
**Color App. Models**	CIE L${}^{*}$a${}^{*}$b${}^{*}$	LLAB	CIECAM
LMS nonlin. sensors	5.89 ± 0.04	5.84 ± 0.02	6.55 ± 0.03
ATD nonlin. sensors	**1.04** ± 0.06	2.00 ± 0.02	1.58 ± 0.05
**Statist. Model**	No adaptation	Von Kries	Webster–Clifford
Infomax SPCA	1.12 ± 0.04	**0.95** ± 0.06	**1.08** ± 0.04

**Table 2 entropy-24-01442-t002:** Differential entropy of color through the networks (in bits). Upper bound for *h* in the 3D cube of size 10 is $h_{\mathrm{unif}}=3\;\log_{2}10=9.97$ bits.

Physiol. Models	No Adaptation	Von Kries	Webster–Clifford
Linear LMS input	−4.97 ± 0.04	−4.01 ± 0.04	−4.01 ± 0.02
Linear ATD channels	−1.68 ± 0.03	−0.71 ± 0.03	−0.72 ± 0.01
Pleistochrome	6.24 ± 0.04	7.64 ± 0.0.3	7.56 ± 0.02
**Color App. Models**	CIE L${}^{*}$a${}^{*}$b${}^{*}$	LLAB	CIECAM
LMS nonlin. sensors	1.93 ± 0.04	−2.18 ± 0.02	−1.70 ± 0.03
ATD nonlin. sensors	4.61 ± 0.02	1.10 ± 0.03	2.74 ± 0.01
**Statist. Model**	No adaptation	Von Kries	Webster–Clifford
Infomax SPCA	**8.70** ± 0.04	**8.82** ± 0.03	**8.71** ± 0.03

**Table 3 entropy-24-01442-t003:** Transferred information (input–output mutual information, in bits). Empirical RBIG upper bound for *I* is 14.1 bits (negligible noise of 0.05% deviation).

Physiol. Models	No Adaptation	Von Kries	Webster–Clifford
Linear LMS input	5.0±0.1	5.1±0.1	5.2±0.1
Linear ATD channels	7.7±0.2	7.6±0.2	7.6±0.1
Pleistochrome	8.58±0.05	8.6±0.1	8.7±0.1
**Color App. Models**	CIE L${}^{*}$a${}^{*}$b${}^{*}$	LLAB	CIECAM
LMS nonlin. sensors	6.6±0.1	5.44±0.03	5.8±0.1
ATD nonlin. sensors	**9.8** ± 0.2	7.8±0.2	**8.8** ± 0.3
**Statist. Model**	No adaptation	Von Kries	Webster–Clifford
Infomax SPCA	**8.9** ± 0.2	8.7±0.3	**8.8** ± 0.3

**Table 4 entropy-24-01442-t004:** Chromatic adaptation (KLD between the D65 and A sets, in bits). Perfect color compensation is represented (by definition) by KLD = 0 bits.

Physiol. Models	No Adaptation	Von Kries	Webster–Clifford
Linear LMS input	5.7 ± 0.1	0.84 ± 0.05	**0.67**± 0.06
Linear ATD channels	3.6 ± 0.3	0.83 ± 0.05	0.78 ± 0.04
Pleistochrome	3.6 ± 0.3	0.7 ± 0.1	0.82 ± 0.07
**Color App. Models**	CIE L${}^{*}$a${}^{*}$b${}^{*}$	LLAB	CIECAM
LMS nonlin. sensors	**0.54** ± 0.07	0.72 ± 0.09	0.73 ± 0.07
ATD nonlin. sensors	**0.55** ± 0.07	0.72 ± 0.07	0.72 ± 0.02
**Statist. Model**	No adaptation	Von Kries	Webster–Clifford
Infomax SPCA	2.2 ± 0.1	1.8 ± 0.1	1.6 ± 0.1

**Table 5 entropy-24-01442-t005:** Gains in available information (ΔI, in bits) due to different features of the models.

	Retinal Adaptation	Opponency	Saturation
Physiol. Models	0.1 ± 0.1	**2.5** ± 0.2	1.0 ± 0.2
	**Retinal Adaptation**	**Opponency + Saturation**	
Color App. Mod.Infomax SPCA	0.9 ± 0.90.0 ± 0.3	**2.8** ± 0.4**3.7** ± 0.2

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
