# Peer review of "Information Flow in Biological Networks for Color Vision"

_entropy, 2022, doi:10.3390/e24101442_

Round 1
Reviewer 1 Report
The paper presents an interesting analysis of the information flow in color vision biological networks.
The paper must be complemented with some additional explanations to make it more understandable to a wider audience.
The introduction can be complemented with the following references:
Lungarella M, Sporns O (2006) Mapping Information Flow in Sensorimotor Networks. PLoS Comput Biol 2(10): e144.
Wang, B., Pourshafeie, A., Zitnik, M. et al. Network enhancement as a general method to denoise weighted biological networks. Nat Commun 9, 3108 (2018).
There are several reference values in the experiments results that are not justified and do not have a supporting reference. For instance, Table 1. "Optimal systems should have T = 0 bits".
Table 2, "Upper bound for H in the common cube is 9.97 bits"
Table 3, "Upper bound for I is 14.1 bits"
What is the meaning in the context of your application of a negative value of the differential Entropy, H, in Table 2?
This paragraph appears at the end of the section 4, experiments and results. Since it is related to preparation of the data, it should go before in the Materials and Methods section "in our experiments measuring I every considered sensor was subject to
Gaussian noise whose standard deviation was 5% of the total deviation of the response. We computed
I between these noisy representations and the input (assuming a negligible noise in the input, with
standard deviation of 0.05% of the total deviation). As convenient reference, I between the noiseless
input and the negligible-noise input (i.e. the maximum available information) is 14.1 0.1 bits."
The Materials and Methods section must be detailed with equations and with more details to reproduce the results of the experiments.
Page 7, has a statement that is not always true: "Regarding indirect measures of efficiency, representations that minimize the redundancy, T,
and maximize the entropy, H, are better for signal representation for the following reasons..".
For instance, for coding purposes a signal with lower entropy can be represented with fewer bits.
Page 8, discussion section needs a supporting reference o more explanation. It is not clear this statement "...linear opponent channels rotate the input LMS representation". The paper does not mention or explain rotations.
Also in the discussion section, this statement is not clear: "... the nonlinear nature of the opponent channels equalizes the responses". It needs more explanation or a supporting reference. You must explain what is the meaning of equalize in this context.
The conclusions section has statements that are not support by the results of the paper or a relevant reference. Examples:
"...they reduce about 75%.."
".. retain about 35% of the chromatic information..."
"...retain about 65% of the information..."
Minor observations:
Please define acronyms before using them.
Some acronyms are not defined and assumed to be known.
Page 2, LSM sensors.
Page 2, KLD
Page 3, The sequence of the building blocks is not shown as a Figure. It only has the (1) symbol, like an equation.
Page 3, sequence of the building blocks. Maybe the notation is clearer if subscripts are used instead of superscripts.
Page 3, a reference is missing.
(RBIG) [? ].
Pag 5, ICA is not defined.
Author Response
"Please see the attachment."

Reviewer 2 Report
In this manuscript, a series of color vision models are analyzed in terms of (a) how the redundancy among the chromatic components is modified along the network, and (b) how much information is transferred from the input data to the noisy response. It is concluded that opponent channels and their nonlinear nature are more important than chromatic adaptation at the retina stage.
The manuscript provides an insight research methodology and direction from the viewpoint of combining color vision, neuroscience and coding theory. It is strongly recommended to accept the article after minor revision. Detailed comments are listed as follows:
[1] (Paragraph 4, page 4) It is necessary to decide several parameters of CIECAM02 model used in the manuscript more detailed, such as factor determining degree of adaptation (F), impact of surrounding (c) and chromatic induction factor (Nc) etc. Please list these parameters of CIECAM02 used in the manuscript. The case of LLAB model is similar. Please describe its parameters.
[2] (Paragraph 4, page 4) Notations mistake. Please revise them:
-- CIE Lab model --> CIE L*a*b* model.
-- L Lab model --> LLAB model.
[3] (Last Line 1, page 3) RBIG: Lack of reference number.
Author Response
"Please see the attachment."

Round 2
Reviewer 1 Report
The author has considered all the recommendations and greatly improved the manuscript.
The analysis of information flow for biological color vision is very interesting.
Page 6, please check the following word:
negentropy.
Do you mean entropy?